# Light-controlled switching of the spin state of iron(III)

Sreejith Shankar [1,2], Morten Peters[1], Kim Steinborn[1], Bahne Krahwinkel[1], Frank D. Sönnichsen[1], Dirk Grote[3], Wolfram Sander [3], Thomas Lohmiller [4,5], Olaf Rüdiger [4] & Rainer Herges[1]

Controlled switching of the spin state of transition metal ions, particularly of $Fe^{II}$ and $Fe^{III}$, is a prerequisite to achieve selectivity, efficiency, and catalysis in a number of metalloenzymes. Here we report on an iron(III) porphyrin with a photochromic axial ligand which, upon irradiation with two different wavelengths reversibly switches its spin state between low-spin ($S = 1/2$) and high-spin ($S = 5/2$) in solution (DMSO-acetone, 2:598). The switching efficiency is 76% at room temperature. The system is neither oxygen nor water sensitive, and no fatigue was observed after more than 1000 switching cycles. Concomitant with the spin-flip is a change in redox potential by ~60 mV. Besides serving as a simple model for the first step of the cytochrome P450 catalytic cycle, the spin switch can be used to switch the spin-lattice relaxation time $T_1$ of the water protons by a factor of 15.

[1] Otto Diels-Institute for Organic Chemistry, University of Kiel, 24118 Kiel, Germany. [2] Photosciences and Photonics Section, Chemical Sciences and Technology Division, CSIR – National Institute for Interdisciplinary Sciences and Technology (CSIR – NIIST), Industrial Estate P.O., Pappanamcode, Thiruvananthapuram 695019, India. [3] Ruhr-Universität Bochum, Organic Chemistry II, 44801 Bochum, Germany. [4] Max-Planck-Institute for Chemical Energy Conversion, 45470 Mülheim a. d. Ruhr, Germany. [5] Present address: Berlin Joint EPR Lab, Institute for Nanospectroscopy, Helmholtz-Zentrum Berlin für Materialien und Energie, Kekuléstraße 5, 12489 Berlin, Germany. Correspondence and requests for materials should be addressed to R.H.(email: rherges@oc.uni-kiel.de)

Spin-state switching is the key step in a number of enzymatic reactions, particularly in C-H activation processes. Controlled spin flips guide reactants and intermediates into reaction pathways with low barriers, and induce catalytic cycles leading to products that otherwise would be formed only under drastic reaction conditions. Cytochrome P450 is a particularly well-investigated example[1,2]. Substrate binding to cytochrome P450 triggers a change in the spin state of the heme-bound Fe(III) from the resting state, low spin ($S = \frac{1}{2}$), to high spin ($S = \frac{5}{2}$). Concomitantly, the reduction potential changes, and a cascade of reactions follows, leading to the selective oxidation of alkyl and aromatic C-H bonds. Methane monooxygenases (MMOs) even convert methane to methanol under ambient conditions[3], a process without precedence in artificial systems. Besides the importance in spin-catalysis, molecular spin-state switching in solution and on surfaces is gaining importance in other fields as well. Molecular spin switches have been proposed as components in molecular electronics, photonics, and spintronics[4]. Particularly interesting, as well, is their potential use as responsive contrast agents in magnetic resonance imaging (MRI), providing spatial information of metabolic parameters in vivo[5].

Whereas spin-state switching in the solid state (spin-crossover mainly in Fe(II) complexes) is a well-investigated phenomenon, the first magnetically bistable compound in solution was published only recently (2011)[6]. Since then, research in the field considerably gained momentum[7–10]. The original molecular spin switches were based on Ni(II) porphyrins[11–13]. The square planar complexes are diamagnetic ($S = 0$), but turn paramagnetic ($S = 1$) upon coordination of axial ligands. Coordination/decoordination from the Ni ion was achieved by the use of photoswitchable axial ligands that were either covalently bound to the porphyrin or substituted in such a way that the cis configuration would not bind because of steric hindrance. A number of further transition metal ions change their spin state upon changing their coordination number, such as $Fe^{2+}$, $Fe^{3+}$, $Co^{2+}$, $Mn^{2+}$, and $Mn^{3+}$, providing larger variations ($\Delta\mu$) in their magnetic moments than $Ni^{2+}$.

Herein, we report the first photoswitchable, magnetically bistable Fe(III) porphyrin in solution that is stable at ambient conditions in solution, including moisture and air. Upon irradiation with light of 365 and 435 nm, the spin-state switches reversibly between low spin ($S = \frac{1}{2}$) and high spin ($S = \frac{5}{2}$), accompanied by a change in redox potential. The species involved and their spin states are characterized using several independent methods, and the mechanism is elucidated in detail.

## Results

**Spectroscopic investigations.** In terms of redox behavior and spin states, iron is probably the most complicated and adaptable element in the periodic table. A number of different spin states are possible in every oxidation state, many of which are close in energy. More than 3000 papers have been published solely on iron porphyrins, particularly in view of their biological relevance. Several reviews summarize the field[14,15]. Iron(III) porphyrins exhibit four different spin states: low spin, $S = \frac{1}{2}$, $(d_{xz}d_{yz})^4(d_{xy})^1$, low spin, $S = \frac{1}{2}$, $(d_{xy})^2(d_{xz}d_{yz})^3$, intermediate spin, $S = \frac{3}{2}$, $(d_{xy})^2(d_{xz},d_{yz})^2(d_{z^2})^1$, and high spin, $S = \frac{5}{2}$, $(d_{xy},d_{xz},d_{yz},d_{z^2},d_{x^2-y^2})$. Moreover, if the energy difference of intermediate spin ($S = \frac{3}{2}$) and high spin ($S = \frac{5}{2}$) is close to the spin-orbit coupling constant (which is mostly the case), linear combinations of high-spin and intermediate-spin states are formed, which have been coined admixed-spin states, $S = a \frac{3}{2} + (1 - a) \frac{5}{2}$ ($a = 0–1$)[16]. The axial ligands and the nature of the porphyrin are the primary determinants of the spin state. Pure 4-coordinate Fe(III) porphyrins without axial coordination are intermediate spin ($S = \frac{3}{2}$)[17]. Axial coordination of two oxygen ligands gives rise to an admixed-spin state ($S = \frac{3}{2}, \frac{5}{2}$) if they are weak, or a high-spin species ($S = \frac{5}{2}$) if they are strong. Fe(III) porphyrins with two strong nitrogen ligands are low spin ($S = \frac{1}{2}$). Besides their electronic properties, the steric bulk of axial ligands is decisive in controlling the spin state. Bulky substituents neighboring the coordinating nitrogen atom not only reduce the binding constant, but also favor the high-spin state[18]. If the counter-ion is weakly binding (e.g., $ClO_4^-$) and the axial ligands are weak, more basic porphyrins (i.e., electron-donating substituents) favor admixed-spin states, and less basic porphyrins (electron-withdrawing substituents) favor low-spin states. Strongly binding anions, such as chloride in the absence of axial ligands, favor the formation of high-spin complexes. Hydrogen bonding to the chloride weakens coordination and the spin-state changes to intermediate spin[19]. Upon judicious choice of the electronic and steric properties of the porphyrin and axial ligands, the complexes can be tuned in such a way that they are close to the $S = \frac{1}{2} \rightleftharpoons S = \frac{5}{2}$ spin-crossover point[20,21]. In this region, small changes in ligand field strength or steric hindrance would switch the spin state of Fe(III). Since spin-state chemistry and axial ligand exchange processes of Fe(III) porphyrin complexes are very intricate, we used several independent methods to characterize our compounds in solution: (a) proton nuclear magnetic resonance ($^1H$ NMR) pyrrole and phenyl shifts, (b) ultraviolet–visible (UV–vis) absorption, (c) magnetic moments, and (d) electron paramagnetic resonance (EPR) spectroscopy. The largest change in spin state ($\Delta S = 2$) in Fe(III) porphyrins is achieved upon switching between high spin ($S = \frac{5}{2}$) and low spin ($S = \frac{1}{2}$). In designing our molecular spin switch, we avoided admixed-spin ($S = \frac{5}{2}, \frac{3}{2}$) and intermediate-spin states ($S = \frac{3}{2}$), not only because they reduce the change in magnetic moment ($\Delta\mu$) but also because these species are sensitive to water and oxygen, and they are difficult to characterize (e.g., the $^1H$ NMR pyrrole shift varies between $+80$ and $-63$ ppm)[14]. To achieve a reliable, robust, and large spin switch, we designed a system which would change the axial coordination between two strong oxygen ligands (high spin, $S = \frac{5}{2}$) and two strong nitrogen ligands (low spin, $S = \frac{1}{2}$) upon irradiation with two different wavelengths.

For our experiments, we chose the readily available Fe(III) tetraphenylporphyrin perchlorate (FeTPPClO$_4$) as the base porphyrin, and photoswitchable azopyridines as axial ligands (photodissociable ligands, PDLs). FeTPPClO$_4$ is admixed-spin ($S = \frac{3}{2}, \frac{5}{2}$) in non-coordinating and weakly coordinating solvents[11,22–24], and very sensitive towards water and oxygen[25]. In acetone, a 6-coordinate complex (Fig. 1) is formed with two weakly binding acetone molecules as axial ligands. The formation constants $K_1'' = 0.82$ and $K_2'' = 1.08$ L mol$^{-1}$ have been determined by following the $^1H$ NMR shifts of the phenyl protons upon titration of a solution of FeTPPClO$_4$ in CD$_2$Cl$_2$ with acetone-$d_6$ (see Supplementary Figures 21, 24, 25, Supplementary Table 4, and Methods, Calculation of apparent equilibrium constants). Pyrrole shift (40.2 ppm), magnetic moment (4.5 B.M., see Methods, Magnetic susceptibility—Evans measurements) and EPR signals (medium-intensity, high-spin signals at $g = 5.94$ and 2.0, and a very small signal at $g = 4.7–4.8$[16] (see Supplementary Figures 36-37 and Methods, EPR spectroscopy) are indicative that the species is admixed spin with a substantial contribution of high spin. To generate a pure high-spin ($S = \frac{5}{2}$) complex, and to increase $\Delta S$ upon switching, we added dimethyl sulfoxide (DMSO)-$d_6$ to the solution which is known to form a 2:1 high-spin ($S = \frac{5}{2}$) complex with FeTPPClO$_4$ (Fig. 1 and Supplementary Figure 1)[26]. High-spin iron(III) porphyrins exhibit a typical Q band at ~690 nm (Supplementary Figure 19a). Upon titration with DMSO and monitoring the increase of the absorption at 686 nm, we

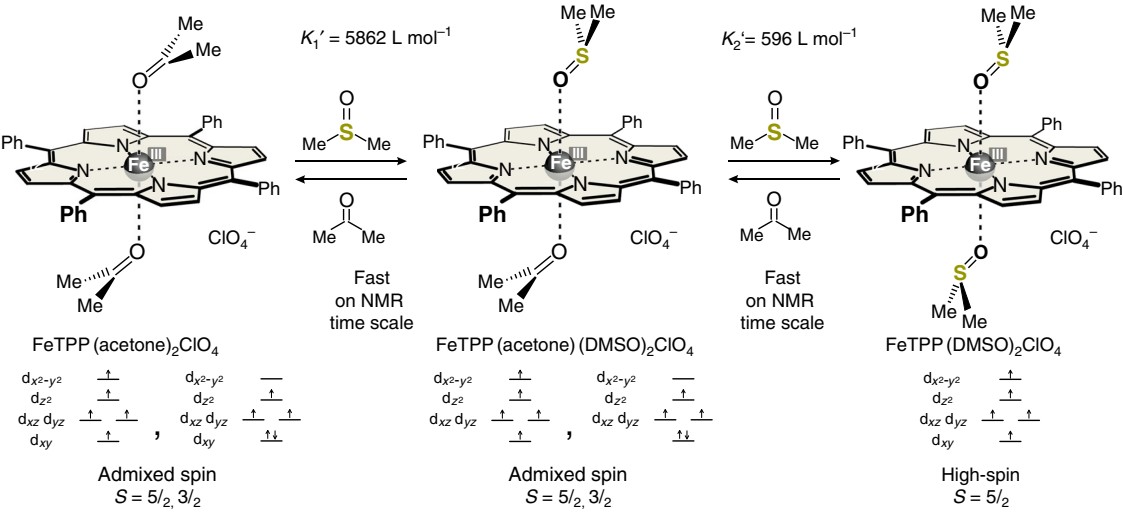

**Fig. 1** Determination of formation constants. Formation constants of high-spin FeTPP(DMSO)$_2$ClO$_4$ determined by following the absorption at 686 nm upon titration of the FeTPP(acetone)2ClO$_4$ complex in acetone with DMSO-$d_6$ at 25 °C. These values were confirmed by NMR titration

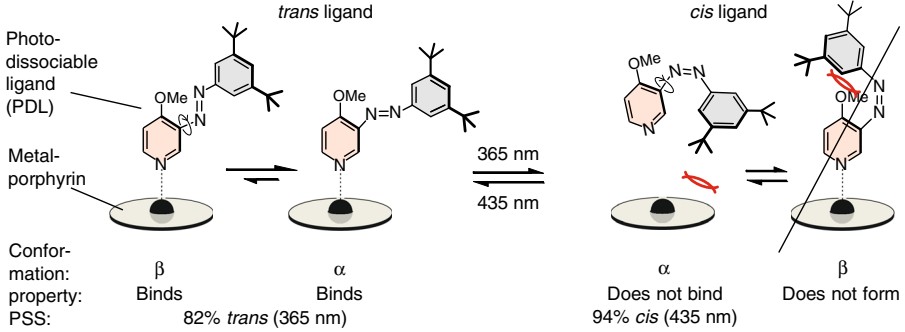

**Fig. 2** Design of the photodissociable ligand (PDL) based on photoswitchable phenyl azopyridine. Irradiation with UV light (365 nm) switches the *trans* isomer to the *cis* form. Visible light triggers reisomerization back to the *trans* form. The photostationary states (PSS) of the two processes are *trans/cis* 75:25 (435 nm), and 6:94 (365 nm). The methoxy group in 4-position of the pyridine was introduced to increase the coordination strength, and to prevent the formation of the *β* conformation of the *cis* isomer. The two *t*-Bu groups at the phenyl ring increase steric repulsion with the porphyrin in the *cis* form, however, have no effect on the coordination of the *trans* configuration

determined the complex formation constants $K_1' = 5862$ L mol$^{-1}$ and $K_2' = 596$ L mol$^{-1}$ (see Supplementary Figures 19b, 26-28). These formation constants were confirmed by NMR titration ($K_1' = 5372$ L mol$^{-1}$ and $K_2' = 580$ L mol$^{-1}$) following the phenyl and pyrrole shifts (see Supplementary Figures 22, 29, 30, Supplementary Table 7, and Methods, Calculation of Apparent equilibrium constants)[27,28].

The magnetic moment of 5.8 B.M. (Evans, see Methods, Magnetic susceptibility—Evans measurements) and the EPR data (large-intensity, high-spin signals at $g = 5.92$ and 2.0, see Supplementary Figure 36 and Methods, Electron paramagnetic resonance (EPR) spectroscopy) confirm a high-spin $S = 5/2$ state for a solution of 0.2 mM FeTPP$^+$ and 47.12 mM DMSO-$d_6$ in acetone-$d_6$. The precise composition of the solution derived from $K_1'$ and $K_2'$ is 0.01% FeTPP(acetone)$_2$$^+$, 3.5% FeTPP(acetone) (DMSO)$^+$, and 96.5% FeTPP(DMSO)$_2$$^+$. This stable solution, mainly consisting of high-spin FeTPP(DMSO)$_2$ClO$_4$, was used for further switching experiments with PDLs based on azopyridine.

**Azopyridine-based PDL.** The azopyridine was designed in such a way that it strongly coordinates to Fe(III) in its *trans* configuration (Fig. 2), forming a low-spin $S = 1/2$ complex. However, upon switching to the *cis* isomer under UV light, steric hindrance

between the *t*-Bu groups and the porphyrin ring prevents binding (for the synthesis of the azopyridine see Supplementary Figure 41 and Methods, Synthesis)[29,30].

Irradiation with visible light (435 nm) regenerates the *trans* isomer which returns to the coordination site. Conversion of the thermodynamically more stable *trans* to the metastable *cis* isomer with UV light is very efficient. The photostationary state (PSS) at 365 nm is 94% *cis*, and 6% *trans*, and the PSS of the back-reaction at 435 nm is 75% *trans*, and 25% *cis* (see Supplementary Figure 5, Supplementary Table 1, and Methods, PSS of the phenyl azopyridine **2** and PSS of azopyridine). This incomplete conversion to the binding *trans* isomer does not compromise the overall switching efficiency, because an excess of the ligand is used[21]. Hence, the azopyridine is ideally suited as a PDL. The PSS are virtually the same in the presence of the porphyrin (Supplementary Figure 6, Supplementary Tables 1, 11, 12). Obviously, no quenching of the excited state of the azopyridine by the porphyrin occurs. Substitution of the pyridine ring in 4-position with a methoxy group was necessary to increase the binding constant, and to prevent the formation of the *β* conformation of the *cis* isomer (see Fig. 2 and Supplementary Figure 4). Thermal half-life of the metastable *cis* isomer with 164 days at 27 °C is very long (Supplementary Figure 7). In the presence of the Fe(III) porphyrin the half-life is reduced to 18 days which is still more than sufficient to serve as a

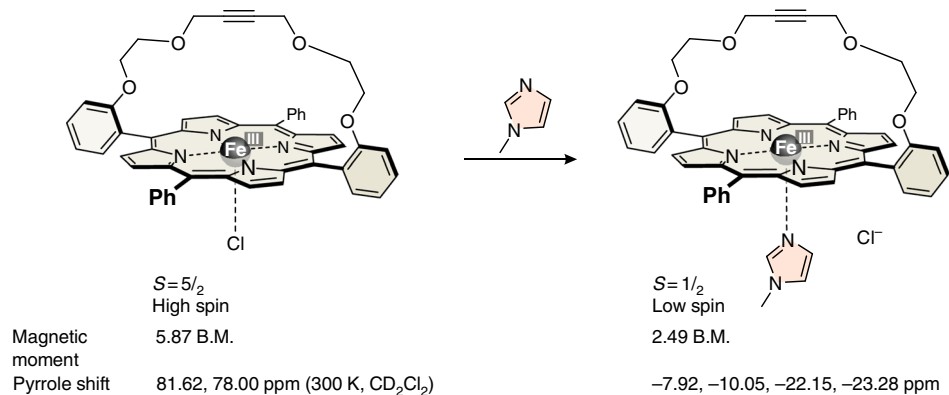

|  |  |  |
|---|---|---|
|  | $S = \frac{5}{2}$ High spin | $S = \frac{1}{2}$ Low spin |
| Magnetic moment | 5.87 B.M. | 2.49 B.M. |
| Pyrrole shift | 81.62, 78.00 ppm (300 K, CD$_2$Cl$_2$) | −7.92, −10.05, −22.15, −23.28 ppm |

**Fig. 3** A strapped porphyrin as a model compound. A strapped porphyrin was used to model the properties of a 5-coordinate Fe(III) porphyrin with one strong axial nitrogen ligand. The [1]H NMR pyrrole shifts and the effective magnetic moment indicate that the chloride complex is high spin and the 1:1 imidazole complex is low spin

photochemical switch (see Supplementary Figure 8 and Methods, Half-life ($t_{\frac{1}{2}}$) of *cis* azopyridine (*cis*-**2**)). With a strong Soret band of the Fe(III) porphyrin around 400 nm (Supplementary Figure 17), the photoexcitation may not be restricted to the PDL ; the event of the Fe(III) porphyrin getting excited, at least partially, cannot be rigorously excluded while irradiating at 365 or 435 nm. The porphyrin in its excited state does not seem to have a noticeable impact on the photoisomerization mechanism of the PDL, since the PSS of the ligand remains largely unaffected by the presence or absence of the porphyrin. However, a slightly longer irradiation time is required for the PDL to arrive at its PSS in the presence of the strongly absorbing porphyrin.

## Discussion

The light-driven spin-state switching experiments with our electronically fine-tuned Fe(III) porphyrin system and the sterically and electronically designed axial ligand were monitored by several methods: NMR, UV–vis, EPR, Evans, and potentiometry, to unanimously determine the structure and magnetic properties of the species involved in the spin-state switching process.

In contrast to acetone/DMSO, axial ligand exchange of DMSO/ azoypyridine is slow on the NMR time scale. Upon titration of the high-spin acetone/DMSO solution of FeTPPClO$_4$ (0.2 mM FeTPP$^+$, 47.12 mM DMSO-$d_6$ in acetone-$d_6$) with the PDL, the signal of the pyrrole protons at 68.6 ppm decreases and a new signal at −14.7 ppm appears (Supplementary Figure 2). Unfortunately, the high-spin signal at 68.6 ppm is very broad, and integration of the NMR signals is not sufficiently accurate to derive a binding model, and to accurately determine coordination constants. We therefore monitored the titration by UV–vis spectroscopy (Supplementary Figure 20). High-spin ($S = \frac{5}{2}$) FeTPP$^+$ complexes exhibit characteristic bands (Q bands) at 690 nm (Supplementary Figure 19a)[31,32], whereas low-spin ($S = \frac{1}{2}$) FeTPP$^+$ does not absorb above 650 nm (Supplementary Figure 20a). The decreasing absorption at 686 nm as a function of added azopyridine yields a binding isotherm (Supplementary Figure 20b), which was analyzed by non-linear optimization[33] (see Methods, Calculation of apparent equilibrium constants). A binding model that includes all conceivable species should consider the starting complex FeTPP(DMSO)$_2^+$, the mixed complex FeTPP(DMSO)(azopy)$^+$, and the 1:2 complex FeTPP(azopy)$_2^+$. The solution of the initial DMSO complex FeTPP(DMSO)$_2^+$ has been thoroughly characterized (see above), the properties of the pure low-spin 1:2 complex FeTPP(azopy)$_2^+$ can be determined by dissolving FeTPPClO$_4$ in pure 4-methoxypyridine. However, the properties of the mixed complex cannot be directly determined, since it is not a stable species. It is not known whether coordination of a

single strong nitrogen ligand to FeTPP$^+$ is sufficient to induce a spin switch to the low-spin state. We therefore synthesized a strapped porphyrin in which one of the two axial binding sites is sterically shielded (Fig. 3, see Methods, Syntheses). The chloride complex (similar to FeTPPCl) is high spin as indicated by the pyrrole proton shifts (81.6, 78.0 ppm), and by the magnetic moment (5.87 B.M., see Methods, Magnetic susceptibility—Evans measurements). Note that in contrast to FeTPP$^+$, the structure of the strapped porphyrin does not exhibit a fourfold symmetry axis and the eight pyrrole protons are not symmetry equivalent. Upon addition of N-methyl-imidazole, the chloride is replaced by the strong nitrogen ligand. The coordination of a second ligand is prevented by the bridge. Magnetic moment (2.49 B.M.) and the pyrrole proton shifts (−7.92, −10.05, −22.15, and −23.28 ppm) clearly indicate that the complex is low spin ($S = \frac{1}{2}$) with a slow exchange of the ligand on the NMR time scale (see Supplementary Figures 42-43 and Methods, Synthesis). Hence, coordination of one nitrogen ligand is sufficient to induce the spin switch.

Based on the properties of the strapped porphyrin, we assume that the mixed complex FeTPP(DMSO)(azopy)$^+$ is low spin as well. Further evidence for the low-spin state of the mixed complex is provided by the fact that the fitting of the binding isotherm is considerably superior with the assumption that the spin change is induced by the coordination of the first azopyridine ligand (see Supplementary Figures 31–33 and Methods, Calculation of apparent equilibrium constants). Based on this model, both binding constants $K_1$ and $K_2$ can be accurately determined by UV–vis titration of the DMSO complex with the azopyridine (see above). According to the analysis of the binding isotherm, the first DMSO ligand is replaced by azopyridine with $K_1 = 314$ L mol$^{-1}$ and the second with $K_2 = 868$ L mol$^{-1}$ (Fig. 4 and Supplementary Table 10). Thus, azopyridine is a much stronger ligand than DMSO and displacement of the second DMSO is much more favorable than expulsion of the first DMSO ligand. Therefore, the concentration of the mixed (DMSO/azopyridine) complex is always small. This is favorable for our spin switching efficiency, because the isomerization of one azopyridine ligand should facilitate the replacement of the second by DMSO leading to an efficient conversion to the high-spin FeTPP (DMSO)$_2^+$ complex.

Spectroscopic parameters as well as magnetic properties consistently prove that the addition of azopyridine converts the high-spin ($S = \frac{5}{2}$) FeTPP(DMSO)$_2^+$ to a low-spin ($S = \frac{1}{2}$) complex. Upon addition of 75 equivalents of *trans* azopyridine to a solution of 0.1 mM FeTPP$^+$, and 25.87 mM DMSO-$d_6$ in acetone$_6$, the EPR high-spin signals at $g = 5.92$ and 2.0 are reduced to approximately 3% of their initial intensity (see Supplementary

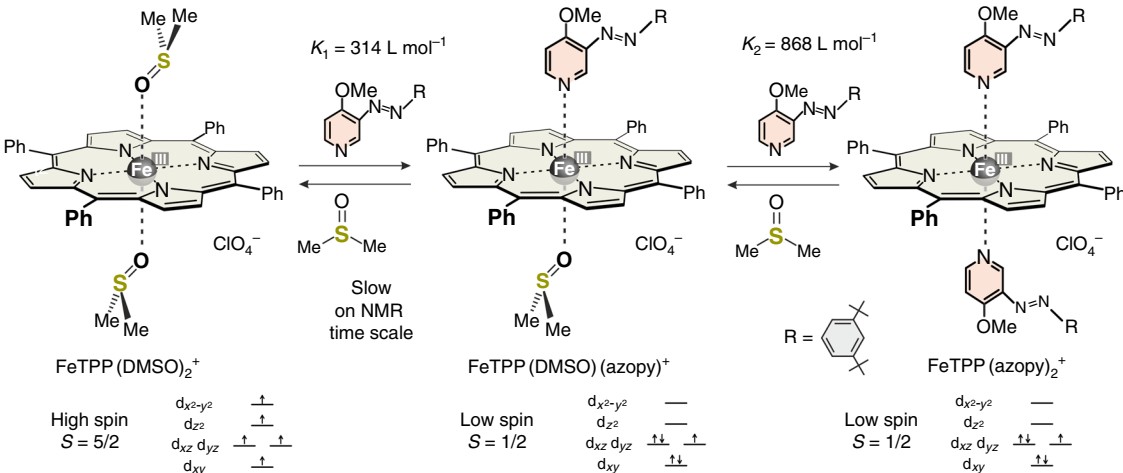

**Fig. 4** Titration of high-spin FeTPP(DMSO)$_2^+$ with the photodissociable ligand (PDL) azopyridine. The binding isotherm was derived from the decrease of the UV–vis absorption band at 686 nm upon addition of the *trans* azopyridine to a solution of 0.1 mM FeTPP$^+$, and 25.87 mM DMSO-$d_6$ in acetone-$d_6$ at 25 °C (see Methods, Far–vis spectroscopy of the spin-state changes and Calculation of apparent equilibrium constants). The binding constants $K_1$ and $K_2$ were determined under the assumption that the mixed complex FeTPP(DMSO)(azopy)$^+$ is low spin

Figure 36 and Methods, EPR spectroscopy), and the UV–vis absorption at 686 nm almost completely vanished (Supplementary Figure 20a). According to Evans measurements (see Methods, Magnetic susceptibility—Evans measurements), the effective magnetic moment of the solution is 2.1 B.M. The high-spin pyrrole proton signal at 68.8 ppm disappeared, and a new signal at −14.7 ppm (typical for low-spin FeTPP$^+$ complexes) appeared (Supplementary Figure 2). Unfortunately, only a very low intensity large $g_{max}$ signal for the low-spin complex at $g = 3.4$ was visible in the EPR spectrum in frozen acetone (Supplementary Figure 36), most probably because of solubility problems, and aggregation. In frozen CH$_2$Cl$_2$, however, the low-spin signal is visible more clearly (Supplementary Figures 37–38). More detailed information about the exact composition of the solution can be derived from the binding constants: $K_1'$, $K_2'$ and $K_1$, $K_2$. In a solution of 0.2 mM FeTPP$^+$, 47.12 mM DMSO-$d_6$, and 15 mM *trans* azopyridine in acetone-$d_6$, the following species are in equilibrium: 0.0006% FeTPP(acetone)$_2^+$, 0.17% FeTPP(acetone)(DMSO)$^+$, 4.7% FeTPP(DMSO)$_2^+$, 11.9% FeTPP(DMSO)(azopy)$^+$, and 83.2% FeTPP(azopy)$_2^+$. The species with at least one strong nitrogen ligand (azopyridine), and that with two strong oxygen ligands (DMSO), add up to 95.1% low-spin and 4.7% high-spin porphyrin, respectively, which is consistent with the EPR observations.

This solution that is predominantly low spin was subjected to irradiation with light of 365 nm. The *trans* azopyridine isomerizes to the *cis* configuration with a conversion rate of 94.4% *cis* (determined by $^1$H NMR, Supplementary Figure 5) in the PSS. Isomerization, in turn, leads to increasing steric demand, and dissociation of the *cis* azopyridine ligand. The binding constant of *cis* azopyridine to FeTPP$^+$ was too low to be determined. Addition of a solution of 94% *cis* and 6% *trans* azopyridine to an acetone-$d_6$ or acetone-$d_6$/DMSO-$d_6$ solution of FeTPP$^+$ does not change the $^1$H NMR spectrum beyond what would be expected from the coordination of the residual *trans* isomer (6%) (see Supplementary Figure 3b and Methods, Switching experiments in NMR,). Hence, *cis* azopyridine is a much weaker ligand than DMSO (in contrast to the *trans* isomer); therefore, DMSO rebinds to the axial coordination sites replacing the azopyridine. The situation is rather complicated since both azopyridine ligands must be replaced by DMSO to switch the spin state. Mixed complexes, such as FeTPP(DMSO)(azopyridine)$^+$ in the equilibrium, have to be considered as well. The exact composition of

the solution in the PSS can be calculated from the binding constants of DMSO ($K_1'$, $K_2'$), and *trans* azopyridine ($K_1$, $K_2$) to FeTPP$^+$ determined by NMR and UV–vis titration (see above), and the *cis*/*trans* ratio of azopyridine at the PSS, determined by $^1$H NMR (Supplementary Figure 6). At the PSS (365 nm) the solution contains 81.4% high-spin FeTPP(DMSO)$_2^+$, 11.3% low-spin FeTPP(azopy)(DMSO)$^+$, and 4.3% low-spin FeTPP(azopy)$_2^+$ (and very small amounts of the acetone complexes, for a detailed list see Table 1). Upon irradiation with light of 435 nm, the *cis* azopyridine returns to its strongly binding *trans* configuration (24.5% *cis*, 75.5% *trans*), and replaces the DMSO ligands. At the PSS (435 nm), 77.2% low-spin FeTPP(azopy)$_2^+$, 14.7% low-spin FeTPP(DMSO)(azopy)$^+$, and 7.8% high-spin FeTPP(DMSO)$_2^+$ are in equilibrium. In essence, upon irradiation with 365 and 435 nm, we are able to switch between 92:8 and 16:81 low-spin/high-spin equilibria. This corresponds to a switching efficiency of 76%, considering the low-spin ratios (Fig. 5, Table 1, see also Methods, Calculation of apparent equilibrium constants).

To investigate the fatigue resistance of the spin switch, we irradiated the above solution with 365 and 435 nm in an alternating sequence and monitored the $^1$H NMR spectrum as a function of the number of switching cycles (Fig. 6a and Supplementary Figure 35b). After 5 min irradiation of the NMR tube with 365 nm (light-emitting diode (LED), 12 × 400 mW), the low-spin signal at −14.7 ppm disappeared, and the broad high-spin signal at 66.8 ppm appeared. Upon irradiation of the NMR tube with 435 nm (LED, 12 × 380 mW), the high-spin signal disappeared and the low-spin signal reappeared. There was no observable change in the $^1$H NMR spectrum after 1000 switching cycles under ambient conditions (air, moisture, 300 K, Supplementary Figure 3a).

Paramagnetic metal ions such as Mn$^{2+}$ ($S = 5/2$) or Gd$^{3+}$ ($S = 7/2$) are known to reduce the longitudinal (or spin–lattice) ($T_1$), and the transverse (spin–spin) relaxation times ($T_2$) of the solvent nuclei[34]. The relaxation mechanism is mainly due to magnetic dipole interactions during coordination of the solvent molecule to the paramagnetic ion (inner sphere relaxation)[35,36]. Upon rapid coordination/decoordination, a single paramagnetic metal ion can relax more than 10$^6$ solvent protons within an NMR or MRI experiment, which finally leads to a reduction of $T_1$ and $T_2$ of the bulk solvent. Switching of the spin state of FeTPP$^+$ between $S = 5/2$ and $S = 1/2$ leads to a switching of the capability to induce nuclear spin relaxation $R_1$ (relaxivity $R_1 = \Delta T_1^{-1} \cdot c^{-1}$,

**Table 1 Photostationary states of the PDL and different possible complexes of FeTPP$^+$**

| | PSS 435 nm (%) | | PSS 365 nm (%) | | |
|---|---|---|---|---|---|
| *trans* azopy | 75.5 | | 5.6 | | |
| *cis* azopy | 24.5 | | 94.4 | | |
| FeTPP(azopy)$_2^+$ | 77.2 | ⟨ 91.9 | 4.4 | ⟨ 15.7 | low-spin |
| FeTPP(azopy)(DMSO)$^+$ | 14.7 | | 11.3 | | low-spin |
| FeTPP(DMSO)$_2^+$ | 7.8 | | 81.4 | | high-spin |
| FeTPP(acetone)(DMSO)$^+$ | 0.3 | | 2.9 | | admixed-spin |
| FeTPP(acetone)$_2^+$ | 0.001 | | 0.01 | | admixed-spin |

At the photostationary state generated upon irradiation with light of 435 nm (PSS 435 nm), the combined concentration of low-spin species is 91.9%, and at PSS 365 nm this concentration decreases to 15.7%. Hence, the switching efficiency is 76.2%

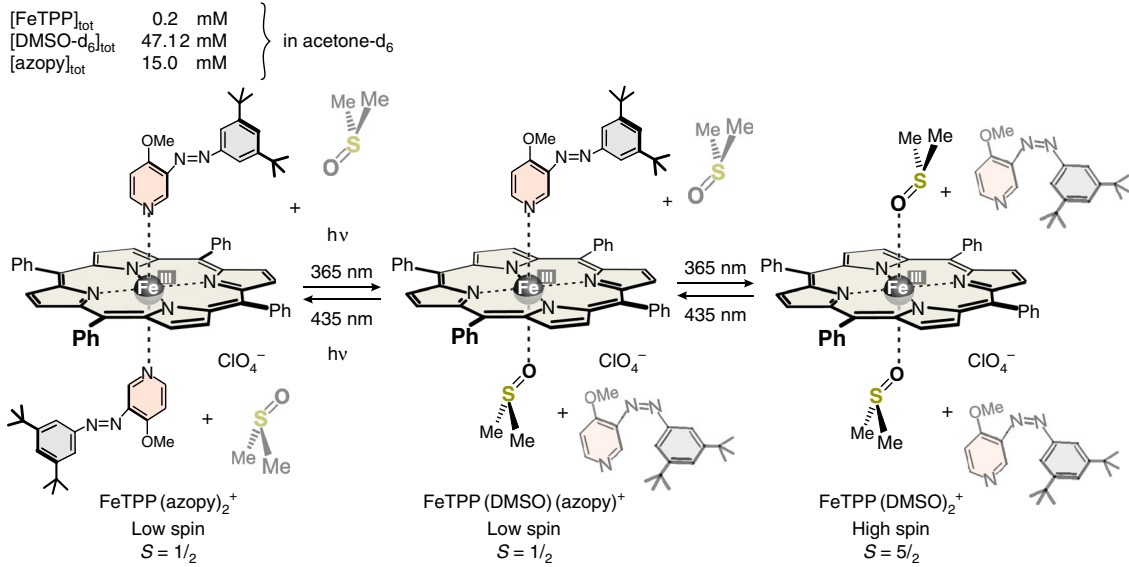

**Fig. 5** Light-induced spin switching of iron(III) porphyrin (FeTPP$^+$) using azopy as a PDL. The composition of the solution is calculated from the binding constants $K_1'$, $K_2'$ and $K_1$, $K_2$, and is listed for the photostationary states (PSS) at 435 and 365 nm in Table 1. The calculations are based on the assumption that *cis* azopyridine does not coordinate to FeTPP$^+$, and that FeTPP(DMSO)(azopy)$^+$ is a low-spin complex

where $\Delta T_1$ is the difference of $T_1$ with and without relaxation agent and $c$ the concentration of the relaxation agent). High-spin FeTPP$^+$ has a larger magnetic moment than low-spin FeTPP$^+$ and therefore exhibits a larger relaxivity. Low-spin FeTPP$^+$ ($S = 1/2$) is not completely diamagnetic; however, both axial coordination sites are occupied by strong nitrogen ligands which prevent a fast exchange with the solvent on the NMR time scale (inner sphere relaxation). The low magnetic moment as well as the blockage of axial coordination sites lead to a considerably smaller relaxivity $R_1$ of the low-spin complex. The relaxivity values $R_1$, and relaxation times $T_1$ of a 2.0 mM solution of the high-spin FeTPP$^+$ complex in an acetone-$d_6$/DMSO-$d_6$ solution containing 1% acetone and 1% water are given in Table 2. The relaxivity $R_1$ of acetone, and water with FeTPP$^+$ in the high-spin state is 12.9, and 17.7 times higher than in the low-spin state. The relaxation time $T_1$ changes from 5.05 to 1.68 s (acetone), and from 0.56 to 0.04 s (water) upon low-spin to high-spin conversion (Table 2, Supplementary Figures 9–11, 13–16, and Supplementary

Table 2). Thus, the efficiency in relaxivity switching is considerably higher than in a previous Ni$^{2+}$ porphyrin-based system[5].

In nature, upon binding of a substrate to cytochrome P450 in its low-spin, 6-coordinate resting state, the axial water ligand is released and a 5-coordinate high-spin Fe(III) porphyrin is formed[2]. Concomitant with the spin flip is a change in redox potential[37]. A similar trend has been observed by Walker et al.[38] who investigated a number of iron porphyrins (as models of cytochromes) including FeTPP$^+$ in *N,N*-dimethylformamide in the presence and absence of pyridine ligands. Switching of the azopyridine ligand from the *trans* to the *cis* configuration should lead to a shift of the reduction potential. Typical cyclic voltammograms (CVs) obtained for the Fe(III)/Fe(II) redox couple for the low-spin FeTPP$^+$ complex before and after irradiation at 365 nm for 20 min are shown in Fig. 6b. The low-spin complex was found to exhibit a well-defined 1$e^-$ oxidation–reduction curve with a half-wave potential of 82±4 mV (vs. standard hydrogen

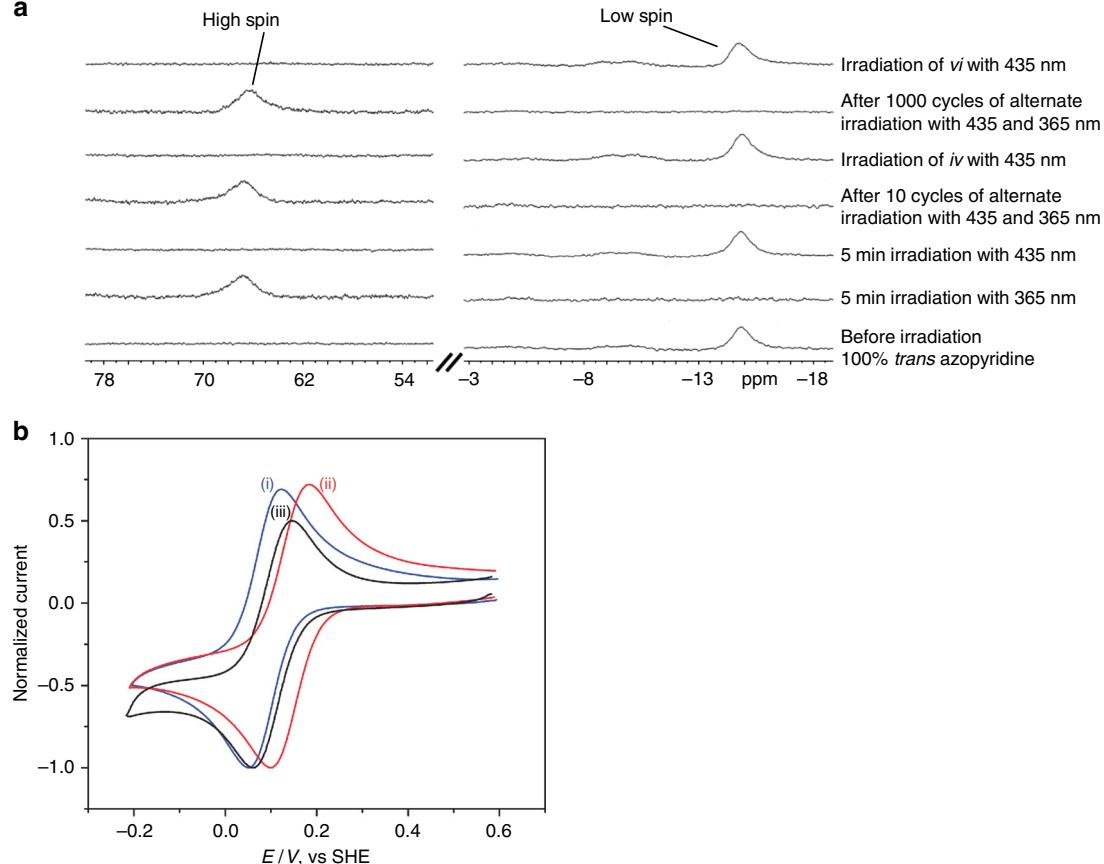

**Fig. 6** Switching of the properties of iron porphyrin FeTPP$^+$. **a** Long-term stability (1000 switching cycles) of a solution of 0.2 mM FeTPPClO$_4$, 47.12 mM DMSO-$d_6$, and 15 mM azopyridine in acetone-$d_6$ at 300 K upon irradiation with 365 and 435 nm in an alternating sequence (left) sections of $^1$H NMR spectra between 54 and 78 ppm (high-spin region) and (right) between −18 and −3 ppm (low-spin region). **b** Light-induced switching of the redox potential of a solution containing 0.2 mM FeTPPClO$_4$, 40 mM tetrabutylammonium perchlorate as the supporting electrolyte and 47.2 mM DMSO in acetone (see Methods, Electrochemical measurements). Blue: cyclic voltammogram before irradiation; red: after irradiating with 365 nm for 20 min; black: after irradiation of the above with 435 nm light for 30 min. The experiment was performed using a platinum disk working electrode, platinum counter electrode, and Ag/AgNO$_3$ 0.01M reference electrode at a scan rate of 50 V/s (see Methods, Electrochemical measurements)

**Table 2 Relaxivities and relaxation times of the solvent protons**

| | | Relaxivity $R_1$ (mM$^{-1}$ s$^{-1}$) | | Relaxation time $T_1$ (s) | |
|---|---|---|---|---|---|
| | | Acetone | Water | Acetone | Water |
| i | High spin | 0.219 | 10.6 | 1.68 | 0.037 |
| ii | Low spin | 0.017 | 0.6 | 5.05 | 0.564 |

Relaxivities $R_1$ and relaxation times $T_1$ of a solution of 2.0 mM FeTPP$^+$ and 47.12 mM DMSO-$d_6$ (i) without and (ii) with 15 mM *trans* azopyridine in acetone-$d_6$, containing 1% acetone and 1% water

potential (SHE)). After 30 min under 435 nm illumination, the Fe (III)/Fe(II) reduction potential almost returns to its value before the 365 nm illumination (Fig. 6b). If the electrochemical experiments are performed strictly under nitrogen, the switching is reversible (see Supplementary Figure 39 and Methods, Electrochemical measurements). Under air obviously the reduced Fe(II) species reacts with oxygen and the reaction becomes irreversible.

This work presents the first light-controlled molecular spin switch based on Fe(III). Starting complex is the readily accessible, admixed-spin ($S = 3/2$, $5/2$) Fe(III) tetraphenylporphyrin perchlorate. In an acetone/DMSO solution it forms a well-defined high-spin ($S = 5/2$) complex with two axial DMSO ligands. The spin switching process is induced by a photoswitchable azopyridine ligand. This ligand is designed in such a way that it coordinates to the iron porphyrin in its *trans* configuration with a binding constant 180 times stronger than DMSO, forming a low-spin ($S = 1/2$) complex with two axial *trans* azopyridine ligands. Upon irradiation with 365 nm, the *trans* isomer of the azopyridine ligand photoisomerizes to the *cis* configuration with a conversion ratio of 94.4%. The *cis* isomer is sterically hindered and does not bind to the iron porphyrin, and thus is replaced by DMSO, regenerating the high-spin state. This ligand exchange/spin switching process is reversible. Light (435 nm) converts the *cis* azopyridine back to the *trans* form, which again replaces the DMSO ligands. No fatigue or side reactions have been observed after more than 1000 switching cycles under air and moisture at room temperature.

Alongside the spin-state flipping, several properties change reversibly. The capability of reducing the NMR proton relaxation time (relaxivity) changes by a factor of more than 10. Properly designed iron(III) porphyrins could therefore be used as functional contrast agents in MRI[5,6]. Upon switching the spin state from $S = 1/2$ to $S = 5/2$, the redox potential shifts to more positive values. The active center of cytochrome P450 contains a low spin Fe(III) porphyrin in its resting state. The iron(III) center undergoes a spin flip upon substrate binding, leading to a change in redox potential, which in turn triggers a cascade of reactions,

which finally leads to selective C-H oxidation[1,2]. Spin state switching is also a crucial step in the conversion of methane to methanol by methanotrophic bacteria, using $Fe^{3+}$-containing enzymes (MMOs)[39]. Nature obviously uses spin flipping to solve difficult catalytic problems that are not susceptible to simple transition state lowering on a given spin-state energy hypersurface.

Control of the spin state might as well be used as a key step in a number of particularly obstinate cases of catalysis in artificial systems, or even industrial processes. We believe that our results could provide a contribution towards this end.

## Methods

**General experimental procedure**. All reactions were carried out in hot air-dried glassware with magnetic stirring under nitrogen atmosphere (when required) using commercially available reagent-grade solvents (dried when necessary, but without purification), and all evaporations were carried out under reduced pressure on Büchi rotary evaporator or Heidolph rotary evaporator below 50 °C, unless otherwise noted. Yields refer to chromatographically and spectroscopically homogeneous materials, unless otherwise stated. Most reagents were purchased from Sigma-Aldrich, ABCR, Alfa Aesar, or Merck and were used as received. Solvents (reagent grade) were purchased from Sigma -Aldrich, ABCR, and Merck. Specific experimental conditions are provided under each section below.

The high-resolution (HR) mass spectra were measured with an APEX 3 FT-ICR with a 7.05 T magnet by co. Bruker Daltonics. Electron impact and matrix-assisted laser desorption/ionization (MALDI) mass spectra were measured with a Biflex III by co. Bruker.

For column chromatography purifications silica gel (Merck, particle size 0.040–0.063 mm) was used. $R_f$ values were determined by thin layer chromatography on Polygram® Sil G/UV$_{254}$ (Macherey–Nagel, 0.2 mm particle size).

Infrared spectra were measured on a Perkin-Elmer 1600 Series FT-IR spectrometer with an A531-G Golden-Gate-Diamond-ATR-unit. Signals were abbreviated with w, m, and s for weak, medium, and strong intensities. Broad signals were additionally labeled with br.

$^1$H and $^{13}$C{$^1$H} NMR spectra were recorded on Bruker DRX 500 ($^1$H NMR: 500 MHz, carbon-13 nuclear magnetic resonance ($^{13}$C NMR): 125 MHz) and Bruker AV 600 ($^1$H NMR: 600 MHz, $^{13}$C NMR: 150 MHz) instruments. Chemical shifts are expressed in ppm ($\delta$), using tetramethylsilane (TMS) as internal standard for $^1$H and $^{13}$C nuclei ($\delta H = 0$; $\delta C = 0$). Multiplicities of NMR signals are designated as s (singlet), d (doublet), t (triplet), q (quartet), quin (quintet), b (broad), and m (multiplet, for unresolved lines). $^{13}$C NMR spectra were recorded with complete proton decoupling, on above-mentioned spectrometers.

NMR spectra were measured in deuterated solvents (Deutero). The degree of deuteration is given within parentheses. $^1$H NMR spectra in reference to the following signals:

$$\text{Chloroform} - d_1 (99.8\%) : \delta = 7.26 \, \text{ppm(s)}.$$

$$\text{Dichloromethane} - d_2 (99.6\%) : \delta = 5.32 \, \text{ppm(t)}.$$

$$\text{Acetone} - d_6 (>99.5\%) : \delta = 2.05 \, \text{ppm(quin)}.$$

**NMR investigations**. Long-term $^1$H NMR switching experiments were performed on a Bruker 500 MHz NMR spectrometer in acetone-$d_6$ (>99.5% d). To a 0.2 mM solution of FeTPPClO$_4$ (**1**) in acetone-$d_6$ (523 µL) in an NMR tube, DMSO-$d_6$ was added (2 µL), followed by a 120 mM acetone-$d_6$ solution of *trans* azopyridine (75 µL, final concentration of 15 mM). The NMR tube was sealed and shaken well before measurement. The $^1$H NMR spectrum showed a characteristic low-spin signal at −14.72 ppm. The tube was then irradiated for 5 min with light of wavelength 365 nm using a custom-made LED machine (12 × 400 mW). During irradiation, constant mixing was ensured via rotating the tube by a mechanical rotator with air cooling using an electric ventilator (fan). The NMR tube was then transferred to the spectrometer and the corresponding spectrum showed the characteristic high-spin signal at +66.8 ppm. The first cycle of irradiation was completed by irradiating the sealed tube with light of wavelength 435 nm (LED 12 × 380 mW) for another 5 min using the exact same set up as described above. The partial switching experiments were repeated for 1000 cycles. Each cycle consisted of the photoconversion of *cis* azopyridine to *trans* azopyridine via alternate irradiation using lights of two different wavelengths (365, 435 nm) and measurement after regular intervals. The LEDs were connected to an automated timer device and the on and off times were fixed at a constant values of 5 min each for each wavelength. The solution was stable even after 1000 continuous cycles of partial switching.

PSS of the phenyl azopyridine **2**: A 15 mM solution of the azopyridine in acetone-$d_6$ (598 µL) containing DMSO-$d_6$ (2 µL) in a sealed NMR tube was kept at 60 °C for 24 h. The $^1$H NMR spectra of this sample confirmed 100% *trans* azopyridine. The sample was then irradiated for 2, 7, and 120 min with a light of wavelength 365 nm using the exact same set up as described above. $^1$H NMR spectra of the sample irradiated for 2, 5, and 10 min showed negligible changes in the concentration of *cis* azopyridine, suggesting that the photostationary equilibrium (94.4 ± 0.3% *cis*, 5.6 ± 0.3% *trans*) was reached within 2 min. Similar irradiation of the sample with light of wavelength 435 nm again resulted in a photostationary equilibrium (24.6 ± 0.3% *cis*, 75.4 ± 0.3% *trans*) as shown in Supplementary Table 1. The PSS was not greatly affected by the presence of the porphyrin in the system.

Half-life ($t_{\frac{1}{2}}$) of *cis* Azopyridine (*cis*-**2**): *Cis* azopyridines, in general, usually exhibit a similar half-life as the corresponding azobenzenes. The half-life ($t_{\frac{1}{2}}$) of *cis* azopyridine (*cis*-**2**) was measured using $^1$H NMR. *Cis*-**2** at its photostationary equilibrium (94.4 ± 0.3% *cis*, 5.6 ± 0.3% *trans*) in a sealed NMR tube was measured for several days in a 500 MHz NMR spectrometer. The half-life ($t_{\frac{1}{2}}$) of *cis*-**2** was found to be extremely high (>164 days, Supplementary Figure 7) at room temperature in acetone-$d_6$ or acetone-$d_6$/DMSO-$d_6$, whereas the same in presence of the paramagnetic Fe(III) porphyrin was significantly lower (~18 days, Supplementary Figure 8).

Magnetic susceptibility—Evans measurements: The paramagnetic susceptibility of admixed-spin complex FeTPP(acetone)$_2$$^+$, high-spin and low-spin complexes FeTPP(DMSO)$_2$$^+$ and FeTPP(azopy)$_2$$^+$ were determined via the standard Evans measurements using $^1$H NMR spectroscopy. An NMR tube with a coaxial insert, both sealable, was used. For the admixed-spin complex FeTPP(acetone)$_2$$^+$, the outer tube was filled with a 0.2 mM solution of porphyrin FeTPPClO$_4$ (**1**) in acetone-$d_6$ (530 µL) and the inner tube was filled with a 0.2 mM solution of diamagnetic ZnTPP (zinc tetraphenylporphyrin) in acetone-$d_6$ (60 µL). ZnTPP that is completely diamagnetic compensates the diamagnetic contribution of the porphyrin macrocycle to the observed magnetic susceptibility. For the high-spin complex FeTPP(DMSO)$_2$$^+$, the outer tube was filled with a 0.2 mM solution of porphyrin FeTPPClO$_4$ in acetone-$d_6$–DMSO-$d_6$ (530 µL, 598:2 v v$^{-1}$) and the inner tube was filled with a 0.2 mM solution of diamagnetic ZnTPP in acetone-$d_6$–DMSO-$d_6$ (60 µL, 598:2 v v$^{-1}$). For the low-spin complex FeTPP(azopy)$_2$$^+$, the outer tube was filled with a 0.2 mM solution of porphyrin FeTPPClO$_4$ containing 15 mM azopyridine in acetone-$d_6$–DMSO-$d_6$ (530 µL, 598:2 v v$^{-1}$) and the inner tube was filled with a 0.2 mM solution of diamagnetic ZnTPP containing 15 mM azopyridine in acetone-$d_6$–DMSO-$d_6$ (60 µL, 598:2 v v$^{-1}$). For the strapped high-spin complex **17**, the outer tube was filled with a 0.721 mM solution of strapped porphyrin **17** in dichloromethane-$d_2$ (530 µL) and the inner tube was filled with a 0.721 mM solution of diamagnetic ZnTPP in dichloromethane-$d_2$ (60 µL). For the strapped low-spin complex **18**, the outer tube was filled with a 1.63 mM solution of strapped porphyrin **18** containing 66.8 mM 1-methylimidazole in dichloromethane-$d_2$ (530 µL) and the inner tube was filled with a 1.63 mM solution of diamagnetic ZnTPP containing 1-methylimidazole (66.8 mM) in dichloromethane-$d_2$ (60 µL). The concentration of the internal standard, TMS, was maintained constant in the inner and outer tubes. $^1$H NMR spectra were recorded in a 500 MHz spectrometer at a constant temperature of 300 K. The sample (inner and outer tubes) was allowed to equilibrate at this temperature for at least 15 min before measurement. The following Eq. (1) was used to calculate the paramagnetic susceptibility from the experimentally measured shift in TMS signals between the inner and outer tubes:

$$\chi_M = \frac{3\delta f M}{4\pi f m} + \chi_M^0 + \frac{\chi_M^0 (d_0 - d_x)}{m} \chi_{dia}^M, \tag{1}$$

where $\chi_M$ is the molar paramagnetic susceptibility in (cm$^3$ mol$^{-1}$), $\delta f$ is the frequency difference between the TMS peaks of the inner and outer tube (Hz), $M$ the molecular weight of the substance (g mol$^{-1}$), $f$ the frequency of the NMR instrument (Hz), $m$ the mass of the substance in 1 mL of solution (g), $\chi^0_M$ the mass susceptibility of the solvent (cm$^3$ mol$^{-1}$), $d_0$ the density of the solvent (g cm$^{-3}$), $d_x$ the density of the solution (g m$^{-3}$), and $\chi^M_{dia}$ the diamagnetic correction to the magnetic susceptibility (cm$^3$ mol$^{-1}$).

Since all the solutions used in this experiment are dilute, the density of the solvent and the solutions may be considered equal, thereby nullifying the factor ($d_0 - d_x$). The diamagnetic correction ($\chi^M_{dia}$) has been taken care of by using equivalent diamagnetic components in the inner tube. Since the same solvents are used in the inner and outer tubes, the solvent correction factor $\chi^0$ could also be neglected. Thus, the equation for calculating the molar paramagnetic susceptibility effectively reduces to Eq. (2):

$$\chi_M = \frac{3\delta f M}{4\pi f m}. \tag{2}$$

From the known values of molar paramagnetic susceptibility, the corresponding magnetic moment ($\mu_{eff}$) may be obtained via Eq. (3):

$$\mu_{eff} = 2.828\sqrt{\chi_M T}, \tag{3}$$

where $T$ is the temperature (K).

For a 0.2 mM solution of the high-spin complex FeTPP(DMSO)$_2$$^+$, $\delta f = 5.78$ Hz, $\chi_M = 0.0138$ cm$^3$ mol$^{-1}$, and $\mu_{eff} = 5.76$ B.M.

For a 0.2 mM solution of low-spin complex FeTPP(azopy)$_2$$^+$, $\delta f = 0.78$ Hz, $\chi_M = 0.00186$ cm$^3$ mol$^{-1}$, $\mu_{eff} = 2.11$ B.M.

For a 0.2 mM solution of admixed-spin complex FeTPP(acetone)$_2$$^+$, $\delta f = 3.53$ Hz, $\chi_M = 0.00843$ cm$^3$ mol$^{-1}$, and $\mu_{eff} = 4.50$ B.M.

For a 0.721 mM solution of 5,15-strapped iron(III) porphyrin chloride **17**, $\delta f = 24.8$ Hz, $\chi_M = 0.01434$ cm$^3$ mol$^{-1}$, and $\mu_{eff} = 5.87$ B.M.

For a 1.63 mM solution of 5, 15-strapped iron(III) porphyrin methylimidazole **18**, $\delta f = 8.79$ Hz, $\chi_M = 0.00258$ cm$^3$ mol$^{-1}$, and $\mu_{eff} = 2.49$ B.M.

Relaxivity measurements: The relaxation time of acetone and water were determined via NMR spectroscopy (Bruker AC 200) in acetone-$d_6$ + DMSO-$d_6$ + 1% acetone. The longitudinal (or spin-lattice) relaxation time ($T_1$) of acetone and water were obtained by an inversion recovery pulse sequence. The integral of the acetone and water signals were observed as a function of the delay time (Supplementary Figure 8). See Supplementary Table 3 for the calculated values of $T_1$.

The transverse (or spin–spin) relaxation time $T_2$ was determined by a spin echo pulse sequence. The integral of the DMSO signal was observed as a function of the spin echo ($n$) with an echo time ($\tau$) of 10 ms (Supplementary Figure 12). The efficiency of paramagnetic ions in shortening the relaxation time of solvent protons may be determined based on relaxivity ($R_1$ and $R_2$). The plot of the relaxation rate ($1/T_1$ or $1/T_2$) vs. the concentration of the paramagnetic species shows a linear relation and the slope is defined as relaxivity, $R_1$ and $R_2$.

**UV–vis and far–vis spectroscopy investigations.** UV–vis and far–vis absorption spectra were recorded on a Perkin-Elmer Lambda-14 spectrophotometer using quartz cells of 1 cm path length. Spectrophotometric grade solvents (2.0 mL) were employed in optical spectroscopic measurements. Irradiation experiments were performed in acetone containing small amounts of DMSO-$d_6$ (25.87 mM). The temperature during every measurement was fixed at 25 °C using a water-flow system connected to a thermostat.

PSS of azopyridine **2**: Fifty micromolar and 1.0 mM solution of the azopyridine in acetone/DMSO (2 mL, 25.87 mM) in sealed quartz cells were irradiated for 2 min with a light of wavelength 365 nm, with stirring. The UV–vis spectrum was recorded. Extended irradiation (up to 10 min) showed negligible changes in the spectrum of *cis* azopyridine (predominant species), suggesting that the photostationary equilibrium was reached within 2 min. Similar irradiation of the sample with a light of wavelength 435 nm again resulted in a photostationary equilibrium as shown in Supplementary Figure 18. See Supplementary Table 1 for the *cis/trans* ratio in the PSS determined by $^1$H NMR.

Far–vis spectroscopy of the spin-state changes: To a solution of FeTPP(acetone)$_2$$^+$ at room temperature, increasing amounts of DMSO-$d_6$ was added and shaken well. The final concentration of the porphyrin was fixed at 0.1 mM and the final volume at 2.0 mL. The optical spectra were recorded for each addition of DMSO-$d_6$, 3 min after placing the quartz cells in the measurement chamber for temperature equilibration. Near isosbestic point was observed around 626 nm, suggesting transformation from FeTPP(acetone)$_2$$^+$ to FeTPP(DMSO)$_2$$^+$, with intermediates having the similar optical characteristics as either of the species in solution. The process was followed by plotting the corresponding change in absorption at 686 nm at each concentration of DMSO added—the intensity increased with successive addition of DMSO at least until a final DMSO concentration of 10 mM and levelled off thereafter.

To a solution of porphyrin FeTPP(acetone)$_2$$^+$ at room temperature containing DMSO-$d_6$ (25.87 mM), increasing amounts of *trans* azopyridine were added and shaken well. The final concentration of the porphyrin was fixed at 0.1 mM and the final volume at 2.0 mL. The optical spectra were recorded for each addition of *trans* azopyridine, 3 min after placing the quartz cells in the measurement chamber for temperature equilibration. An isosbestic point was observed around 621 nm, suggesting a transformation from FeTPP(DMSO)$_2$ClO$_4$ to FeTPP(azopy)$_2$ClO$_4$, without any intermediates or with intermediates having the same optical characteristics as either of the species in solution. The process was followed by plotting the corresponding change in absorption at 686 nm at each concentration of *trans* azopyridine added—the intensity was found to decrease with successive addition of azopyridine at least until a final concentration of 7 mM and levelled off thereafter.

Switching experiments in optical spectroscopy: To a 0.1 mM solution of FeTPPClO$_4$ in acetone, containing DMSO-$d_6$ (25.87 mM) in a sealed quartz cell, *trans* azopyridine (trans-**2**) (75 eq., 7.5 mM) was added. The quartz cell was sealed and shaken well before acquiring the optical spectrum in the visible region (500–800 nm). The cell was then irradiated for 2 min with a light of wavelength 365 nm, under magnetic stirring and the optical spectrum was obtained. The first cycle of irradiation was completed by irradiating the sealed cell with a light of wavelength 435 nm for another 3 min. The partial switching experiments were repeated for 10 cycles. Each cycle consisted of the photoconversion of *cis* azopyridine to *trans* azopyridine via alternate irradiation using lights of two different wavelengths (365, 435 nm) and measurement after regular intervals. Supplementary Figure 35 right shows reversible changes in absorption at 686 nm as a function of number of irradiation cycles.

**Calculation of apparent equilibrium constants.** The optical changes accompanying the spin-state changes were used to derive the corresponding apparent equilibrium constants. The model reactions as shown in Supplementary Figure 23 were assumed and validated using the excel tool for equilibrium speciation (Equilibrium Speciation Tool)[33] based on Newton–Raphson method and the reported hybrid generic algorithm, in combination with Excel's Solver.

To determine the relevant species in acetone, the change in the phenyl shifts of the porphyrin, upon addition of increasing amounts of acetone-$d_6$ to a solution of 0.2 mM FeTPPClO$_4$ in CD$_2$Cl$_2$, was analyzed, using binding models for both 1:1 and 2:1 complexation. The pyrrole protons were not visible over the complete range of the titration, and thus could not be used for analysis (see Supplementary Figure 21 and Supplementary Table 4).

Formation of FeTPP(acetone)$_2$$^+$ via FeTPP(acetone)ClO$_4$ with $K_1'' = 0.865$ L mol$^{-1}$ and $K_2'' = 1.077$ L mol$^{-1}$ is the most likely model. The observed and calculated shifts are given in Supplementary Table 4 and are depicted in Supplementary Figure 24. The composition of the corresponding solutions is given in Supplementary Table 5 and shown in Supplementary Figure 25. Based on these results the main component in pure acetone-$d_6$ (13600 mM) is complex FeTPP (acetone)$_2$$^+$ (~93%, see supplementary Table 5)[31].

Absorption changes for the titration of a 0.1 mM solution of FeTPPClO$_4$ in acetone with DMSO-$d_6$ (see Supplementary Figure 19) were analyzed using both 1:1 and 2:1 binding models as well as different assumptions for the absorption of the intermediate species FeTPP(acetone)(DMSO)$^+$ in the 2:1 binding model.

Several binding models have been tested and their fitting has been compared: (a) No intermediate (mixed) complex is formed during ligand exchange (very strong cooperativity of ligand exchange). (b) Formation of only the mixed complex (no double ligand exchange, which is very unlikely because DMSO is known to form 2:1 complexes with FeTPPClO$_4$), (c) absorption of the mixed complex FeTPP (acetone)(DMSO)$^+$ is identical to the absorption of FeTPP(acetone)$_2$$^+$, (d) absorption of the mixed complex FeTPP(acetone)(DMSO)$^+$ is identical to the absorption of FeTPP(DMSO)$_2$$^+$, (e) the absorption of the mixed complex is in between FeTPP(acetone)$_2$$^+$ and FeTPP(DMSO)$_2$$^+$. The SSR (sum of squared residuals) values are (a) and (b) $1.5 \times 10^{-3}$, (c) $6.2 \times 10^{-4}$, (d) $8.1 \times 10^{-5}$, and (e) $2.9 \times 10^{-5}$. This suggests formation of an intermediate species FeTPP(acetone) (DMSO)$^+$ according to model (d) or (e). Model (e) is in accordance with the course of the absorption shown in Supplementary Figure 19, which shows a nearly isosbestic point at 621 nm but a different behavior above 750 nm.

In Supplementary Table 6, the composition of the species in solution, and the observed and calculated absorption at 686 nm are given. The speciation is shown in Supplementary Figure 27. Supplementary Figure 26 shows the comparison of observed and calculated absorption. Apparent binding constants are given in Supplementary Table 10.

With the binding constants obtained from the UV–vis experiments ($K_1' = 5862$ L mol$^{-1}$, $K_2' = 596$ L mol$^{-1}$), we were able to fit the shifts of the phenyl and pyrrole protons of FeTPPClO$_4$. These shifts were followed upon subsequent addition of DMSO-$d_6$ to a solution of FeTPPClO$_4$ in acetone-$d_6$ with a fixed concentration of 0.2 mM porphyrin and a fixed total volume of 600 μL (see Supplementary Tables 7-8 and Supplementary Figure 22). Fitting the NMR data with different initial assumptions for the association constants gave similar results. The best fit was obtained for $K_1' = 5372$ L mol$^{-1}$ and $K_2' = 580$ L mol$^{-1}$ ($\Delta_{SSR} = 4 \times 10^{-5}$ (sum of squared residuals)), which results in a difference of 8.4% and 2.8%, respectively. This close agreement can be interpreted as an additional confirmation of the validity of our model.

For the titration of a solution of FeTPPClO$_4$ (0.1 mM) in acetone containing DMSO-$d_6$ (25.87 mM) with *trans* azopyridine, the binding isotherm was analyzed considering both single or double coordination of azopyridine (see Supplementary Table 9). Occurrence of an isosbestic point at 621 nm suggested either clean transformation to a single product or formation of an intermediate FeTPP(DMSO) (azopy)$^+$ with the same optical characteristics as either species in solution (see Supplementary Figure 20). The most likely model was stepwise coordination of two azopyridines, assuming that FeTPP(DMSO)(azopy)$^+$ has the same absorption as the final complex (FeTPP(azopy)$_2$). This implicates that the first coordination of azopyridine induces spin change from high spin to low spin.

Titration with *trans* azopyridine was also followed by $^1$H NMR spectroscopy, but neither the pyrrole shift (slow exchange, broad signals) nor the phenyl shifts (not observable because of excess of azopyridine) could be analyzed.

A refined binding model could be obtained by fitting the data of the titration with *trans* azopyridine by taking into account the apparent binding constants for DMSO-$d_6$ (see Supplementary Table 10). This provided the basis to calculate the concentration of all relevant species in solution for the single titration points and for the switching experiments (see above).

Using the combined model, we were able to determine the composition of the species in the solution of the NMR switching experiments. Assuming that the binding constants of *cis* azopyridine are much lower than those of DMSO-$d_6$ and, consequently, that *cis* azopyridine does not bind, we determined the composition of the solutions in the PSSs after irradiation with 365 and 435 nm, respectively (see Supplementary Tables 11 and 12). Further evidence that *cis* azopyridine is a very weak ligand, or does not bind at all, is provided in Supplementary Figure 3b. With these values, the switching efficiency could be calculated as 76.3% (see Supplementary Figure 34).

**EPR spectroscopy**. Sample solutions of Fe(III) porphyrin complexes were vacuum-sealed in EPR quartz-glass tubes. X-band ($\approx$9.5 GHz) continuous wave (CW) EPR experiments on samples in acetone at 8 K were performed with a Bruker ESP 380E spectrometer equipped with an Oxford Instruments Ltd. ITC liquid He flow system and temperature controller. X-band CW EPR spectra on samples in $CH_2Cl_2$ were recorded at 4.8 K using a Bruker ELEXSYS E500 spectrometer equipped with an Oxford Instruments Ltd. ESR 900 liquid He flow cryostat and an ITC503 temperature controller. All spectra presented were baseline corrected by subtraction of a background spectrum of the resonator with an empty sample tube.

X-band EPR experiments at liquid He temperatures were performed on the various Fe(III) porphyrin complexes dissolved in acetone or $CH_2Cl_2$ to verify their spin states. Supplementary Figure 36 contains spectra of the following complexes of FeTPP$^+$ at a concentration of 0.2 mM with acetone as the solvent: FeTPP (acetone)$_2^+$ (**1a**), FeTPP(DMSO)$_2^+$ (**2**), FeTPP(azopy)$_2^+$ (**3**), FeTPP(MeOPy)$_2^+$ (**4**). **1a**, **2**, and **3** each exhibit a peak around $g = 6$ of different intensity together with a smaller, sharper peak at $g \approx 2$, indicative of either high-spin $S = 5/2$ states ($g_\perp \approx 6$, $g_\parallel \approx 2$) or admixed-spin $S = 3/2$, 5/2 states[40]. As expected, **2** shows the most intense high-spin signal, while in **3**, its intensity is much smaller ($\sim$3% of **2**). **1a** exhibits a $g \approx 6$ signal broader than the one from **2**, of intermediate intensity, and there possibly is another, small feature around $g = 4.7$–4.8. This appearance indicates a contribution from an intermediate $S = 3/2$ spin state and thus an admixed $S = 3/2$, 5/2 state for **1a**. **4** does not contain any $S = 5/2$ component, as expected, but also no other strong signals. In the spectra of **3** and **4**, there could however be weak signals around $g = 3.3$ and $g = 3.1$, respectively, of very small intensity. These $g$ values are characteristic for the so-called "large $g_{max}$" signals[41,42], originating from low-spin $S = 1/2$ states, as expected for these complexes, of $(d_{xy})^2$ $(d_{xz}, d_{yz})^3$ electronic ground state configuration.

In pulse mode, in contrast, no signals from the FeTPPClO$_4$ complexes in acetone could be observed. The reason for this is thought to be the bad glassing properties of acetone, possibly promoting agglomeration of the complexes. Magnetic interactions between the iron centers and concomitant enhanced relaxation rates prevent that electron spin echoes can be detected, while their CW signals can still be measured.

Hence, dichloromethane ($CH_2Cl_2$), which possesses more favorable glassing properties, was chosen as solvent for further experiments. Supplementary Figure 37 shows X-band CW EPR spectra of complexes **3** and **4** (0.2 mM in $CH_2Cl_2$). As in the samples containing acetone, **3** exhibits a high-spin signal at $g = 5.99$, while **4** does not. Both samples show large $g_{max}$ signals of a low-spin $S = 1/2$ state around $g = 3.4$. In Supplementary Figure 38, the EPR spectra of **3** and **4** in samples with acetone or $CH_2Cl_2$ as the solvent are compared. Samples with $CH_2Cl_2$ as the solvent did show also EPR signals in pulse mode (not shown).

Illumination of **3** with UV light of 365 nm wavelength at room temperature results in a $\sim$20-fold increase of the component at $g = 5.99$ (Supplementary Figure 38), consistent with an increase of the fraction of $S = 5/2$ complexes by low-spin to high-spin conversion. However, there is no concomitant decrease of the large $g_{max}$ low-spin signal, which would be expected as a result of such a process. Hence, there is a significant overall increase of EPR signal intensity. Further illumination at room temperature with blue light of 435 nm wavelength of this sample exposed to UV light before did not lead to a decrease of the $g = 5.99$ signal; thus, not indicating a back-conversion process from high spin to low spin that has been observed before for this type of sample. These results from the photoswitching experiments are not quite consistent with the light-induced behavior observed in the corresponding NMR, Evans, and UV–vis experiments at room temperature. Especially, the increase of overall signal intensity upon UV irradiation suggests that not all the Fe(III) ions present in the sample contribute to the measured EPR spectra. The reason is thought to be non-ideal complex solvation in the frozen samples, possibly resulting in cluster formation, even when using $CH_2Cl_2$ as the solvent.

**Electrochemical measurements**. The electrochemical measurements were performed on an Autolab PGSTAT204 potentiostat equipped with a 3-electrode setup. A platinum disk (5 mm diameter) was used as the working electrode, a platinum wire was used as the counter electrode and a Ag/AgNO$_3$ 0.01 M in acetonitrile was used as the reference electrode. The reference electrode was separated from the main cell chamber with a frit on a side-arm filled with supporting electrolyte in acetone. The CVs were obtained in acetone at a porphyrin concentration of 0.2 mM. Tetrabutylammonium perchlorate (TBAP, 40 mM) was used as the supporting electrolyte. The experiments were conducted inside a glovebox at room temperature at a scan rate of 50 mV s$^{-1}$. All measurements were repeated at least three times to obtain reliable potential values. All the potentials are referred to the SHE. The potentials were corrected using hydroxymethyl ferrocene (Fc-MeOH) as an internal reference. At the end of each measurement, 0.2 mg of Fc-MeOH was added to the solution and the midpotential was determined. The reported value of 420 mV vs. SHE was used for the conversion. Upon irradiation of the solution of Fe (III)porphyrin, DMSO, *trans* azopyridine, and TBAP (concentrations see Supplementary Figure 39a), the reduction peak shifts towards a more positive potential, whereas a shift to a more negative potential would be expected if the *cis* azopyridine would not interfere with the electrochemical process. We attribute this behavior to complexation of the reduced Fe(II) species with *cis* azopyridine. Further implications arise from the presence of high concentrations of the supporting electrolyte

(TBAP), which is required for the electrochemical experiment. The electrolyte may perturb the equilibrium between the different spin states since it has been shown to favor the admixed state of porphyrins[3]. Independent experiments towards the elucidation of the chemistry of the Fe(II) species applying NMR, magnetic measurements, EPR and UV are currently underway and shall be reported in due course.

**Syntheses**. Synthesis of porphyrin **1**: Tetraphenylporphyrin and FeTPPCl were synthesized as reported[43,44]. FeTPPClO$_4$ (**1**) was prepared via a modified literature method[45]. The toluene complex obtained after crystallization from toluene was dissolved in dichromethane (purified over basic alumina) and the solvent was removed in vacuo. This procedure was repeated several times until no toluene signals were visible in $^1$H NMR.

HR masses of FeTPPClO$_4$ (**1**) with different ligands (see Supplementary Figure 40):

(a)  acetone: 784.24952 (calc.), 784.24891 (found) for C$_{50}$H$_{40}$N$_4$O$_2$Fe,
(b)  4-methoxypyridine: 886.27132 (calc.), 886.27059 (found) for C$_{56}$H$_{42}$N$_6$O$_2$Fe,
(c)  DMSO: 822.17801 (calc.), 822.17609 (found) for C$_{48}$H$_{38}$N$_4$O$_2$FeS$_2$.

Synthesis of azopyridine **2**, general strategy: The azopyridine **2** was synthesized in three steps from commercially available 3,5-di-*tert*-butylaniline **5** and 3-amino-4-chloropyridine **8** as shown in Supplementary Figure 41. Oxidation of the aniline **5** to the corresponding nitrosobenzene **6** was achieved using oxone® in a mixture of water and $CH_2Cl_2$. Base-mediated coupling of the nitrosobenzene **6** with the aminopyridine **8** afforded the azo-product **9**, which on dechlorination–methoxylation resulted in the azopyridine **2**.

Synthesis of nitrosobenzene **6**[46]: To a solution of Oxone® (12.0 g, 39.0 mmol) in water (100 mL) was added a solution of 3,5-di-*tert*-butylaniline **5** (2.00 g, 9.74 mmol) in $CH_2Cl_2$ (40 mL) and the resulting mixture was stirred at room temperature for 5 h. The formation of the nitroso compound was evident by a change in color of the solution to light green. After 5 h, the phases were separated and the aqueous phase was extracted with $CH_2Cl_2$ (2 × 25 mL). The organic phases were combined, dried over anhydrous MgSO$_4$, and evaporated under reduced pressure. The temperature during evaporation was maintained at 30 °C. The crude solid thus obtained was purified by column chromatography (silica gel, 2:5 $CH_2Cl_2$/ pentane as eluent) to afford the nitroso compound **6** as a light green solid. Fourteen percent of the corresponding nitro compound **7** was also isolated as a yellow powder (1.84 g, 8.34 mmol, 84%). $^1$H NMR (600 MHz, 300 K, CDCl$_3$, TMS) $\delta = 7.82$ (t, $^4J_{4,2} = 1.8$ Hz, 1H, *H*-4), 7.78 (d, $^4J_{2,4} = 1.8$ Hz, 2H, *H*-2), 1.40 (s, 18H, C (C*H*$_3$)$_3$) ppm; $^{13}$C NMR (150 MHz, 300 K, CDCl$_3$, TMS): $\delta = 167.2$ (*C*-1), 152.4 (*C*-3), 129.7 (*C*-4), 115.9 (*C*-2), 35.1 (*C*(CH$_3$)$_3$), 31.3 (C(*C*H$_3$)$_3$) ppm; HRMS (ESI) 219.16231 (calc.), 219.16191 (found) for C$_{14}$H$_{21}$N$_1$O$_1$.

Synthesis of azo-compound **9**: To a solution of the aminopyridine **8** (1.00 g, 7.78 mmol) in pyridine (25 mL), 60% KOH in water (75 mL) was added and the resulting mixture was heated to 80 °C. A solution of the nitrosobenzene **6** (1.80 g, 8.19 mmol) in pyridine (50 mL) was added dropwise over 15 min. The reaction mixture was then heated and kept at 80 °C for 8 h with vigorous stirring. After cooling to room temperature, the phases were separated. The aqueous phase was washed with ethyl acetate (2 × 50 mL). The organic phases were combined, dried over anhydrous MgSO$_4$, and evaporated in vacuo. The crude product was purified by column chromatography (1:4 ethyl acetate/cyclohexane as eluent) to obtain the azo-compound **9** as an orange solid (1.38 g, 4.18 mmol, 54%); m.p.: 93 °C; $^1$H NMR (500 MHz, 300 K, CDCl$_3$, TMS): *trans* isomer: $\delta = 8.77$ (s, 1H, *H*-2), 8.54 (d, $^3J_{6,5} = 5.3$ Hz, 1H, *H*-6), 7.84 (d, $^4J_{8,10} = 1.8$ Hz, 2H, *H*-8), 7.64 (t, $^4J_{10,8} = 1.8$ Hz, 1H, *H*-10), 7.52 (d, $^3J_{5,6} = 5.3$ Hz, 1H, *H*-5), 1.41 (s, 18H, C(C*H*$_3$)$_3$) ppm; *cis* isomer: $\delta = 8.24$ (d, $^3J_{6,5} = 5.3$ Hz, 1H, *H*-6), 7.57 (s, 1H, *H*-2), 7.36 (d, $^3J_{5,6} = 5.3$ Hz, 1H, *H*-5), 7.23 (t, $^4J_{10,8} = 1.8$ Hz, 1H, *H*-10), 6.75 (d, $^4J_{10,8} = 1.8$ Hz, 2H, *H*-8), 1.17 (s, 18H, C(C*H*$_3$)$_3$) ppm; $^{13}$C NMR (125 MHz, CDCl$_3$, 300 K, TMS): *trans* isomer: $\delta = 152.7$ (*C*-7), 152.1 (*C*-9), 150.7 (*C*-6), 144.8 (*C*-3), 143.3 (*C*-4), 139.4 (*C*-2), 126.7 (*C*-10), 125.4 (*C*-5), 118.0 (*C*-8), 35.1 (*C*(CH$_3$)$_3$), 31.2 (C(*C*H$_3$)$_3$) ppm; *cis* isomer: $\delta = 153.2$ (*C*-7), 151.9 (*C*-9), 147.8 (*C*-6), 148.8 (*C*-3), 135.8 (*C*-4), 139.4 (*C*-2), 122.7 (*C*-10), 124.5 (*C*-5), 115.0 (*C*-8), 34.9 (*C*(CH$_3$)$_3$), 31.1 (C(*C*H$_3$)$_3$) ppm; HRMS (ESI): 329.16587 (calc.), 329.16492 (found) for C$_{19}$H$_{24}$N$_3$Cl); FT-IR (film): $\tilde{\nu} = 2963$ (s), 2903 (w), 2866 (w), 1602 (m), 1562 (s), 1460 (m), 1361 (m), 1247 (m), 1187 (m), 1162 (m), 1087 (m), 882 (m), 840 (s), 746 (s), 697 (s), 680 (s) cm$^{-1}$; UV–vis (toluene): $\lambda_{max} = 344$ nm, log $\varepsilon = 4.233$ L mol$^{-1}$ cm$^{-1}$.

Synthesis of azopyridine **2**: Metallic sodium (3.60 g) was dissolved in MeOH (100 mL) under cooling by an ice bath. The resulting solution was added slowly to a MeOH solution (20 mL) of the azo-compound **9** (1.63 g, 4.94 mmol, the azo-compound **9** was not completely soluble in MeOH). The resulting mixture was heated to 50 °C for 2 h with vigorous stirring and was then allowed to cool to room temperature. Stirring was then continued for another 8 h and the solvent was removed under reduced pressure. Crushed ice was added to the residue and was extracted with $CH_2Cl_2$ (3 × 50 mL). The combined organic extracts were dried over anhydrous MgSO$_4$ and the solvent was removed under reduced pressure. The crude product thus obtained was purified by column chromatography (silica gel, 2:1 ethyl acetate/cyclohexane as eluent) to give the azopyridine **2** as a dark orange solid (1.46 g, 4.49 mmol, 91%); m.p.: 100 °C; $^1$H NMR (500 MHz, 300 K, CDCl$_3$, TMS): *trans* isomer: $\delta = 8.62$ (s, 1H, *H*-2), 8.54 (d $\approx$ s, 1H, *H*-6), 7.75 (d, $^4J_{8,10} = 1.8$ Hz, 2H, *H*-8), 7.58 (t, $^4J_{10,8} = 1.8$ Hz, 1H, *H*-10), 7.02 (d, $^3J_{5,6} = 5.7$ Hz, 1 H, *H*-5), 4.06

(s, 3H, O$CH_3$), 1.39 (s, 18 H, C($CH_3$)$_3$) ppm; *cis* isomer: $\delta$ = 8.23 (d, $^3J_{6,5}$ = 5.7 Hz, 1H, *H*-6), 7.78 (s, 1H, *H*-2), 7.20 (t, $^4J_{10,8}$ = 1.8 Hz, 1H, *H*-10), 6.74 (s, 1H, *H*-5), 6.73 (d, $^4J_{8,10}$ = 1.8 Hz, 2H, *H*-8), 3.73 (s, 3H, O$CH_3$), 1.17 (s, 18H, C($CH_3$)$_3$) ppm; $^{13}$C NMR (125 MHz, CDCl$_3$, 300 K, TMS): *trans* isomer: $\delta$ = 161.7 (*C*-4), 153.2 (*C*-7), 152.4 (*C*-6), 152.2 (*C*-9), 139.2 (*C*-2), 138.9 (*C*-3), 126.0 (*C*-10), 117.7 (*C*-8), 108.0 (*C*-5), 56.3 (O$CH_3$), 35.2 (*C*($CH_3$)$_3$), 31.6 (C($CH_3$)$_3$) ppm; *cis* isomer: $\delta$ = 154.9 (*C*-4), 153.9 (*C*-7), 151.4 (*C*-9), 149.6 (*C*-6), 140.7 (*C*-3), 140.5 (*C*-2), 121.9 (*C*-10), 114.4 (*C*-8), 106.7 (*C*-5), 55.4 (O$CH_3$), 34.8 (*C*($CH_3$)$_3$), 31.2 (C($CH_3$)$_3$) ppm; MS (MALDI-TOF): *m/z* = 326.06 [M-H]$^+$; HRMS (ESI): 325.21541 (calc.), 325.21520 (found) for C$_{20}$H$_{27}$N$_3$O); FT-IR (film): $\tilde{\nu}$ = 2953 (w), 2869 (w), 1602 (m), 1581 (s), 1480 (m), 1361 (m), 1272 (s), 1193 (s), 1022 (s), 883 (s), 805 (s), 700 (s) cm$^{-1}$; UV–vis (toluene): $\lambda_{max}$ = 344 nm, log $\varepsilon$ = 4.288 L mol$^{-1}$ cm$^{-1}$.

Synthesis of strapped porphyrin **17**, general strategy: The ether bridge **15** was synthesized in two steps from commercially available chemicals with a yield of 24%. The *meso*-phenyl dipyrromethane **14** was synthesized with a yield of 81%. The strapped iron porphyrin **17** was prepared from **14** and **15** in two steps (see Supplementary Figures 42-43).

Synthesis of meso-phenyl dipyrromethane **14**[47]: Pyrrole **13** (24.0 mL, 347 mmol) and benzaldehyde **12** (850 mg, 8.00 mmol) were dissolved under nitrogen atmosphere and stirred for 15 min. TFA (150 μL) was added and the mixture was stirred at room temperature for 25 min. Subsequently, 200 mL of DCM were added and the mixture was washed with a 0.1 M potassium hydroxide solution (120 mL) and twice with water (120 mL). The combined organic layers were dried over anhydrous magnesium sulfate and the solvent was removed under reduced pressure. The crude product was purified by column chromatography (dichloromethane/cyclohexane: triethylamine (1%), 1:1, $R_f$ = 0.61). A colorless solid was obtained (1.44 g, 6.48 mmol, 81%). $^1$H NMR (500 MHz, CDCl$_3$): $\delta$ = 7.90 (br s, 2H, N*H*), 7.37–7.18 (m, 5H, *H*-7, *H*-8, *H*-9), 6.70–6.68 (m, 2H, *H*-4), 6.17–6.15 (m, 2H, *H*-3), 5.93–5.91 (m, 2H, *H*-2), 5.48 (s, 1H, *H*-5) ppm; $^{13}$C NMR (150 MHz, 300 K, CDCl$_3$): $\delta$ = 142.2 (*C*-6), 132.4 (*C*-1), 128.6 (*C*-7) 128.5 (*C*-8), 126.9 (*C*-9), 117.2 (*C*-4), 108.4 (*C*-3), 107.2 (*C*-2), 44.1 (*C*-5) ppm.

Synthesis of 1,4-bis(2-bromoethoxy)-2-butyne **11**[48]: 1,4-Bis(2-hydroxyethoxy)-2-butyne **10** (1.00 g, 5.75 mmol) was dissolved in dry dichloromethane (10 mL) under nitrogen atmosphere. Subsequently, tetrabromomethane (4.20 g, 12.7 mmol) was added, then triphenylphosphine (3.30 g, 12.7 mmol) was dissolved in dry dichloromethane (20 mL) and slowly added dropwise. The reaction mixture was stirred at 0 °C for 90 min and 15 h at room temperature and afterwards poured onto mixture of dichloromethane (50 mL) and distilled water (50 mL). The combined organic layers were separated, dried over anhydrous magnesium sulfate, and the solvent was removed under reduced pressure. The crude product was purified by column chromatography (cyclohexane/ethylacetate, 1:1, $R_f$ = 0.60) to yield **12** as a yellow oil (1.70 g, 5.70 mmol, 99%). $^1$H NMR (500 MHz, 300 K, CDCl$_3$): $\delta$ = 4.27 (s, 4H, *H*-2), 3.85 (t, $^3J$ = 5.5 Hz, 4H, *H*-3), 3.49 (t, $^3J$ = 5.5 Hz, 4 H, *H*-4) ppm; $^{13}$C NMR (150 MHz, 300 K, CDCl$_3$): $\delta$ = 82.3 (*C*-1), 69.7 (*C*-3), 58.5 (*C*-2), 30.0 (*C*-4) ppm.

Synthesis of the bridge **15**: Salicylaldehyde (223 mg, 1.92 mmol) was dissolved in acetonitrile (20 mL) under nitrogen atmosphere. Potassium carbonate (264 mg, 1.92 mmol) and 1,4-bis(2-bromoethoxy)-2-butyne **11** (375 mg, 1.25 μmol) were added and the reaction mixture was stirred for 16 h at 80 °C. The reaction solution was concentrated and was poured onto distilled water (150 mL). The aqueous phase was extracted three times with dichloromethane (50 mL). The combined organic layers were dried over anhydrous magnesium sulfate and the solvent was removed under reduced pressure. The crude product was purified by column chromatography (cyclohexane/ethylacetate, 1:1, $R_f$ = 0.49). A colorless oil was obtained (87.0 mg, 229 μmol, 24%). $^1$H NMR (500 MHz, 300 K, CDCl$_3$): $\delta$ = 10.53 (s, 2H, *H*-1), 7.83 (dd, $^3J$ = 7.6 Hz, $^4J$ = 1.8 Hz, 2H, *H*-7), 7.53 (td, $^3J$ = 8.0 Hz, $^4J$ = 1.8 Hz, 2H, *H*-5), 7.03 (t, $^3J$ = 7.6 Hz, 2H, *H*-6) 6.99 (d, $^3J$ = 8.4 Hz, 2H, *H*-4), 4.31 (s, 4H, *H*-10), 4.27 (t, $^3J$ = 4.7 Hz, 4H, *H*-8), 3.95 (t, $^3J$ = 4.7 Hz, 4H, *H*-9) ppm; $^{13}$C NMR (150 MHz, 300 K, CDCl$_3$): $\delta$ = 189.8 (*C*-1), 161.1 (*C*-3), 135.9 (*C*-5), 128.3 (*C*-7), 125.2 (*C*-2), 121.1 (*C*-6), 112.8 (*C*-4), 82.43 (*C*-11), 68.0 (*C*-8, *C*-9), 58.9 (*C*-10) ppm; HRMS (EI): 382.14275 (calc.), 382.14164 (found) for C$_{22}$H$_{22}$O$_6$; FT-IR (film): $\tilde{u}$ = 2865 (w), 1682 (s), 1597 (s), 1482 (s), 1452 (s), 1395 (m), 1350 (m), 1285 (m), 1241 (m), 1189 (s), 1161 (s), 1100 (s), 1025 (s), 925 (m), 831 (m), 755 (s), 655 (m), 606 (w), 530 (w) cm$^{-1}$.

Synthesis of 5,15-strapped porphyrin **16**: The bridge **15** (375 mg, 983 μmol) and trifluoro boro etherate (13.9 mg, 98.3 μmol) were dissolved in dichloromethane (350 mL) under nitrogen atmosphere. To this solution *meso*-phenyl dipyrromethane **14** (436 mg, 1.96 mmol), dissolved in dichloromethane (50 mL), was added under stirring over a period of 1 h. After stirring for 15 h, *p*-chloranil (504 mg, 2.05 mmol) was added and stirred for 5 h at 40 °C. Then, the solvent was removed under reduced pressure and the crude product was purified by column chromatography (dichloromethane, $R_f$ = 0.54). A purple solid was obtained (80.0 mg, 102 μmol, 10%); m.p.: 372 °C; $^1$H NMR (600 MHz, CDCl$_3$, 300 K): $\delta$ = 8.81 (s, 8H, *H*-13, *H*-14), 8.54 (dd, $^3J$ = 7.2 Hz, $^4J$ = 1.6 Hz, 2H, *H*-9), 8.36 (s, br, 2H, *H-o*-Ph), 8.03 (s, br, 2H, *H-o*-Ph′), 7.82–7.68 (m, 8H, *H*-7, *H-m*-Ph, *H-m*-Ph′, *H-p*-Ph), 7.50 (t, $^3J$ = 7.5 Hz, 2H, *H*-8), 7.07 (d, $^3J$ = 8.2 Hz, 2H, *H*-6), 3.72–3.69 (m, 4H, *H*-4), 2.53–2.48 (m, 4H, *H*-3), 1.06 (s, 4H, *H*-2), −2.59 (s, br, 2H, N*H*) ppm; $^{13}$C NMR (150 MHz, 300 K, CDCl$_3$): $\delta$ = 159.3 (*C*-5), 142.1 (*C*-17), 134.6 (*C-o*-Ph′), 134.4 (*C-o*-Ph), 133.4 (*C*-9), 131.8 (*C*-10), 130.6 (*C*-13, *C*-14), 130.0 (*C*-7), 127.6 (*C-p*-PH), 126.7 (*C-m*-Ph, *C-m*-Ph′), 120.1 (*C*-8), 119.9 (*C*-15), 119.7 (*C*-16), 116.4 (*C*-12), 115.5 (*C*-11), 112.0 (*C*-6), 78.7 (*C*-1), 69.8 (*C*-4), 66.9 (*C*-3), 56.5 (*C*-9) ppm;

MS (MALDI, TOF): *m/z* = 785 [M]$^+$; HRMS (EI): 784.30495 (calc.). 784.30323 (found) for C$_{52}$H$_{40}$N$_4$O$_4$; FT-IR (film): $\tilde{u}$= 2924 (w), 1596 (w), 1471 (m), 1441 (m), 1348 (m), 1284 (m), 1247 (m), 1184 (m), 1112 (m), 965 (s), 796 (s), 728 (s), 698 (s), 658 (m), 579 (m), 519 (m), 408 (s) cm$^{-1}$.

Synthesis of 5,15-strapped iron(III) porphyrin chloride **17**: The 5,15-strapped porphyrin **16** (27.0 mg, 37.2 μmol) and iron(II) chloride tetrahydrate (180 mg, 669 μmol) were dissolved in degassed acetonitrile (30 mL) under nitrogen atmosphere and refluxed for 4 h. Then, the solvent was removed under reduced pressure and the crude product was dissolved in dichloromethane (40 mL), and then washed twice with brine (50 mL) and water (100 mL). The combined organic layers were dried over anhydrous magnesium sulfate and the solvent was removed under reduced pressure. A brown solid was obtained (30.0 mg, 34.3 μmol, 92%); m.p.: 131 °C; $^1$H NMR (500 MHz, CD$_2$Cl$_2$, 300 K): $\delta$ = 81.62 (s, br, 4H, *H*-pyrrole), 78.00 (s, br, 4H, *H*-pyrrole) ppm; MS (MALDI, TOF): *m/z* = 839 [M-Cl]$^+$, 874 [M]$^+$; HRMS (EI): 838.22299 (calc.), 838.22174 (found) for C$_{52}$H$_{38}$FeN$_4$O$_4$; FT-IR (film): $\tilde{u}$= 2921 (w), 1597 (m), 1442 (m), 1336 (m), 1243 (m), 1000 (m), 801 (m), 719 (m), 659 (m), 541 (m), 463 (s), 436 (w), 418 (w), 408 (s) cm$^{-1}$. Due to the highly diluted NMR samples, the large number of quaternary C-atoms and the paramagnetism, $^{13}$C NMR spectroscopy of **17** did not provide sufficient signal intensities. Therefore, the $^{13}$C NMR spectrum was not analyzable.

Synthesis of 5,15-strapped iron(III)porphyrin 1-methylimidazole complex **18**: The 5,15-strapped iron(III) porphyrin chloride **17** (1.42 mg, 1.63 μmol) was dissolved in 400 μL dichloromethane-$d_2$. To this solution 1-methylimidazole (5.48 mg, 66.8 μmol) dissolved in 80 μL of dichloromethane-$d_2$, was added. $^1$H NMR (500 MHz, CD$_2$Cl$_2$, 300 K): $\delta$ = −7.92 (s, 2H, *H*-pyrrole), −10.05 (s, 2H, *H*-pyrrole), −22.15 (s, 2H, *H*-pyrrole), −23.28 (s, 2H, *H*-pyrrole) ppm; HRMS (ESI): 920.27680 (calc.), 920.27580 (found) for C$_{56}$H$_{44}$N$_6$O$_4$Fe. Due to the highly diluted NMR samples, the large number of quaternary C-atoms and the paramagnetism, $^{13}$C NMR spectroscopy of **18** did not provide sufficient signal intensities. Therefore, the $^{13}$C NMR spectrum was not analyzable.

## Data availability

All the data that support the findings of this study are available within the paper and its Supplementary Information files, or from the corresponding author on reasonable request.

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

## Acknowledgements

We gratefully acknowledge support by the DFG via Sonderforschungsbereich 677. S.S. thanks the Alexander von Humboldt Foundation, Germany and DST-SERB, India, respectively, for Humboldt and Ramanujan research fellowships and O.R. for funding by the Max-Planck Society. This work was supported by the Cluster of Excellence RESOLV (EXC 1069) funded by the Deutsche Forschungsgemeinschaft (DFG).

## Author contributions

R.H. and S.S. designed the experiments; S.S. performed the photoswitching, electro-chemistry, and relaxivity experiments; S.S. and B.K. performed titration experiments; K.S. analyzed the binding isotherms; F.D.S. assisted in the NMR data acquisition and inter-pretation; M.P. synthesized and investigated the strapped porphyrin and performed electrochemistry experiments together with S.S. and O.R.; T.L., D.G., and WS measured the EPR spectra; Evans measurements have been performed by B.K., S.S., and M.P.; R.H. supervised the project; and S.S. and R.H. wrote the manuscript.

## Additional information

**Competing interests:** The authors declare no competing interests.

