## [Peer Review File · Nature Communications]

Editorial Note: This manuscript has been previously reviewed at another journal that is not operating a transparent peer review scheme. This document only contains reviewer comments and rebuttal letters for versions considered at Nature Communications. Mentions of prior referee reports have been redacted.

REVIEWERS' COMMENTS:

Reviewer #1 (Remarks to the Author):

The authors have addressed some of my concerns satisfactorily, and with the switchover from [redacted] to Nature Comm., the work is of sufficient novelty to be published.

As before, but with especially now with the larger audience fielded by Nat. Comm., it is essential that the abstract mention explicitly that the results are achieved in solution (mention the solvent). Indeed, moving from spin state switching in a solution to a thin film/heterostructure remains a sizeable challenge.

Reviewer #2 (Remarks to the Author):

I read a previous version of this manuscript submitted to a NPG journal. I had several concerns about characterization of the complexes, references, and presentation. In the revised manuscript, I think the authors have done a good job of addressing the concerns raised. The only thing I would like clarified is the role of the Fe porphyrin complex in the re-isomerization of the photo-dissociable ligand. Figure S16 shows that the complexes (with variable axial ligands and spin states) can absorb strongly at ~400 nm, so excitation at 365 nm and 435 nm (but more so the latter) will excite the complex as well as the azopyridine ligand, and this might impact the isomerization mechanism. Even if it is not known how (or if) the excited states of the Fe complex might impact the isomerization, it should be pointed out in the manuscript that there is some excitation of the complex at both wavelengths chosen--those photons are not strictly interacting only with the PDI. Otherwise, this is an interesting result, and will be of interest to a wide audience.

REVIEWERS' COMMENTS:

Reviewer #1 (Remarks to the Author):

The authors have addressed some of my concerns satisfactorily, and with the switchover from [redacted] to Nature Comm., the work is of sufficient novelty to be published.

As before, but with especially now with the larger audience fielded by Nat. Comm., it is essential that the abstract mention explicitly that the results are achieved in solution (mention the solvent). Indeed, moving from spin state switching in a solution to a thin film/heterostructure remains a sizeable challenge.

done

Reviewer #2 (Remarks to the Author):

I read a previous version of this manuscript submitted to a NPG journal. I had several concerns about characterization of the complexes, references, and presentation. In the revised manuscript, I think the authors have done a good job of addressing the concerns raised. The only thing I would like clarified is the role of the Fe porphyrin complex in the re-isomerization of the photo-dissociable ligand. Figure S16 shows that the complexes (with variable axial ligands and spin states) can absorb strongly at ~400 nm, so excitation at 365 nm and 435 nm (but more so the latter) will excite the complex as well as the azopyridine ligand, and this might impact the isomerization mechanism. Even if it is not known how (or if) the excited states of the Fe complex might impact the isomerization, it should be pointed out in the manuscript that there is some excitation of the complex at both wavelengths chosen--those photons are not strictly interacting only with the PDI. Otherwise, this is an interesting result, and will be of interest to a wide audience.

We included the following sentence to clarify the influence of the porphyrin on the photoisomerization of the azopyridine: "The porphyrin in its excited state does not seem to have a noticeable impact on the photoisomerization mechanism of the PDL, since the photostationary states of the ligand remain largely unaffected by the presence or absence of the porphyrin. However, a slightly longer irradiation time is required for the PDL to arrive at its PSS in presence of the strongly absorbing porphyrin."